Genome-wide association study reveals the advantaged genes regulating japonica rice grain shape traits in northern China

Chen Hongwei 1
Zhang Xue 1
Tian Shujun 1
Gao Hong 1
Sun Jian 2
Pang Xiu 1
Li Xiaowan 1
Li Quanying 1
Xie Wenxiao 1
Wang Lili 1
Liang Chengwei 1 2
Sui Guomin 3
Zheng Wenjing 1 zwj27@126.com
Ma Zuobin 1 mazuo1984@163.com
1 Rice Research Institute of Liaoning Province, Liaoning Academy of Agricultural Sciences , Shenyang , China
2 Rice Research Institute, Shenyang Agricultural University , Shenyang , China
3 Liaoning Academy of Agricultural Sciences , Shenyang , China
Kutlu Imren
Electronic publication date: 2024 Dec 18
Publication date: 2024
Volume: 12
Electronic Location ID: e18746
Received 2024 Aug 5; Accepted 2024 Dec 2
Copyright: © 2024 Chen et al.
Copyright year: 2024
Copyright holder: Chen et al.
License: This is an open access article distributed under the terms of the Creative Commons Attribution License, which permits unrestricted use, distribution, reproduction and adaptation in any medium and for any purpose provided that it is properly attributed. For attribution, the original author(s), title, publication source (PeerJ) and either DOI or URL of the article must be cited.
License URL: https://creativecommons.org/licenses/by/4.0/

Keywords: Japonica, Grain shape, GWAS, QTLs, Candidate genes

Funding: Shenyang Seed Industry Innovation Project 22-318-2-17 Liaoning Province Agriculture Major Project 2022JH1/10200003 Applied Basic Research Project of Liaoning Province 2022JH2/101300283 This research was supported by the Shenyang Seed Industry Innovation Project (No. 22-318-2-17), the Liaoning Province Agriculture Major Project (No. 2022JH1/10200003) and the Applied Basic Research Project of Liaoning Province (No. 2022JH2/101300283). The funders had no role in study design, data collection and analysis, decision to publish, or preparation of the manuscript.

==============================
Background

Rice, a staple food for over half of the global population, exhibits significant diversity in grain shape characteristics, which impact not only appearance and milling quality but also grain weight and yield. Identifying genes and loci underlying these traits is crucial for improving rice breeding programs. Previous studies have identified multiple quantitative trait loci (QTLs) and genes regulating grain length, width, and length-width ratio; however, further investigation is necessary to elucidate their regulatory pathways and their practical application in crop improvement.

Methods

This study employed a genome-wide association study (GWAS) on 280 japonica rice varieties from northern China to decipher the genetic basis of grain shape traits. Phenotyping included measurements of 11 grain-related traits, such as grain length, width, and area, along with their brown and white rice counterparts. High-density single nucleotide polymorphism (SNP) markers (33,579) were utilized for genotyping, and GWAS was performed using a mixed linear model (MLM) incorporating principal component analysis (PCA) and kinship (K) matrix to account for population structure and relatedness.

Results

Our analysis detected 15 QTLs associated with the 11 grain shape traits, of which five major QTL clusters emerged as crucial. Candidate genes, including LOC_Os01g50720 (qGL1), OsMKK4 (LOC_Os02g54600, influencing qBA2, qWL2, and qWA2), GW5 (LOC_Os05g09520, controlling qGW5, qBW5, qBR5, qWW5, and qWR5), GW6a (LOC_Os06g44100, associated with qGW6, qBW6, qBR6, qWW6, and qWR6), and FZP (LOC_Os07g47330, linked to qWL7), were identified based on functional annotations and haplotype analysis. These findings offer valuable insights into the genetic mechanisms underlying rice grain shape and suggest promising targets for marker-assisted selection to enhance rice quality and yield.

Introduction

The genetic mechanism of rice grain shape is a key field in the study of rice quality and yield (Li et al., 2022a). The characteristics that define rice grain shape, specifically grain length, grain width, and grain length-width ratio, not only play a pivotal role in the appearance and milling quality of rice but also serve as determinant factors influencing grain weight (Yang et al., 2021). Therefore, analyzing the genetic mechanism of rice grain shape, cloning the genes related to grain shape, and establishing the regulatory framework of grain shape formation provide important theoretical bases and gene resources for molecular breeding and high yields of rice, which are conducive to improving rice yield and quality (Xia et al., 2017). Based on the ratio of length to width in brown rice, rice grains can be categorized into short, medium, and long grains. The criteria for classification are as follows: grains with a ratio of ≥3.4 are considered long, those with a ratio of ≥2.3 but <3.4 are classified as medium, and grains with a ratio of <2.3 are designated as short (Awad-Allah et al., 2022). Among them, the long-grain varieties are becoming increasingly favored due to their aesthetically pleasing appearance and their reduced chalkiness in milled rice (Singh et al., 2017).

There are many ways to regulate rice grain shape, including ubiquitin proteases, hormones, epigenetic modifications, etc., which lead to the changes in grain shape through the change of cell number or cell volume (Meng et al., 2022). Many genes and QTLs controlling grain length have been cloned, such as qGL3, SMOS1, GS3, GS2, PGL1, TGW6, PGL2, GL7 and others (Fan et al., 2006; Zhang et al., 2012; Ishimaru et al., 2013; Duan et al., 2014; Kabir & Nonhebel, 2021). Among them, qGL3 encodes a protein phosphatase OsPPKL1 containing two Kelch functional domains, which plays the role of negative regulator in the regulation of rice grain length, and the Kelch functional domain is sufficient and necessary in the negative regulatory function of OsPPKL1 (Zhang et al., 2012; Gao et al., 2019). SMOS1 encodes a transcription factor in rice, that includes an AP2 domain, which regulates grain size through the expansion of cell number and the regulation of cell size (Qiao et al., 2017; Hirano et al., 2017). TGW6 encodes an IAA-glucose hydrolase, which influences the number of grain cells and the length of grain by regulating the supply of IAA (Ishimaru et al., 2013; Kabir & Nonhebel, 2021). GL7 encodes a LONGIFOLIA protein, which is highly homologous to the LONGIFOLIA protein of Arabidopsis thaliana. The increase of GL7 expression can increase the longitudinal cell division of grain and reduce the transverse cell division, resulting in an increase in grain length (Wang et al., 2015a, 2015b). GS3, located in the centromere region of chromosome 3, is a major gene controlling grain length (Fan et al., 2006; Kan et al., 2022). The G-protein-γ subunit encoded by GS3 contains a plant-specific organ size regulation (OSR) domain, the deletion of which leads to increased grain length in rice (Mao et al., 2010). In addition, GW2, GS5, GW5, GS6 and GW8 were the major genes that had been cloned to affect grain width. Among them, GW2 encodes a RING-type protein with E3 ubiquitin ligase activity, which can function in the ubiquitin-proteasome pathway to degrade proteins. The loss or decrease of GW2 expression can lead to the increase of the number of rice glume cells, which in turn increases the grain width, the grain filling speed, the grain weight and the yield of rice (Song et al., 2007; Hao et al., 2021). GW5 encodes a nuclear localization protein that inhibits the activity of glycogen synthase kinase GSK2, thereby regulating the expression level of brassinolide response genes and grain growth response (Weng et al., 2008; Liu et al., 2017). Through the study of GW5 genes in different cultivated rice varieties, it was found that the grain width increased due to the deletion of gene fragments under the action of both artificial and natural selection in the process of rice domestication (Liu et al., 2017). GS5 is a major gene on chromosome 5 that affects grain width, filling and grain weight, and positively regulates rice grain length by encoding serine carboxypeptidase (Li et al., 2011; Xu et al., 2015).

At present, QTL research on grain shape related traits of rice has been widely carried out, mainly to improve the appearance quality of rice, but only a few QTLs have been applied in current breeding programs. Further characterization of QTLs associated with rice grain shape and the cloning of the underlying genes is essential. Additionally, a deeper understanding of the regulatory pathways governed by these genes is necessary to fully exploit their potential in rice breeding programs aimed at improving grain shape traits. With the development of high-throughput sequencing technology, high density SNP markers covering the whole genome have made genome-wide association studies widely used. In the past 10 years, the genetic research of plant quantitative traits based on GWAS has made great progress. GLW7 represents a pivotal rice grain type regulatory gene that has been successfully cloned through the application of GWAS (Si et al., 2016). Lv et al. (2019) conducted a genome-wide association analysis on grain shape-related traits, including grain length, grain width, thousand-grain mass, and length-to-width ratio, using 161 indica rice varieties along with 16,352 SNPs, and identified a total of 38 significant loci. Feng et al. (2016), on the other hand, utilized 469 rice varieties combined with 5,291 SNPs to detect 11 novel QTLs through genome-wide association analysis. Grain shape is not only one of the determining factors of yield traits, but also affects the appearance and milling quality of rice.

In this study, 280 japonica rice varieties from northern China were used to conduct genome-wide association study for grain-shape-related traits, and to explore the relevant loci or genes controlling grain shape. These results will provide a theoretical basis for elucidating the genetic mechanism of rice grain shape and breeding excellent new rice varieties with excellent grain shape.

Materials and methods

Plant materials

The association panel consisted of 280 japonica accessions from northern China (Table S1). These accessions were planted in the experimental field of Liaoning Rice Research Institute (41°N, 123°E) from April to October 2021. The sowing, transplanting and harvesting dates were April 25, May 20 and October 10, respectively. Each accession was planted in a plot of three rows, with eight plants in each row at a spacing of 16.7 × 30 cm, and this was replicated twice. Fertilizer for the cultivated land was 180 kg N ha−1, 60 kg P2O5 ha−1, and 120 kg K2O ha−1, with attention to weeding and pest control.

Phenotyping

When the accessions were mature, five plants were selected in the middle of the plot for harvesting and placed in a cool place to allow the seeds dry naturally. A seed moisture meter is used to detect the grain moisture content, and about 200 grains were selected when it has been reduced to 12–14%. For brown rice, the outer husk is removed, retaining the bran layer and germ. To produce white rice, further milling removes the bran layer and germ using a CLS JNM-1 rice husker. Grain length (GL), grain width (GW), grain length-width ratio (GR), brown rice length (BL), brown rice width (BW), brown rice length-width ratio (BR), brown rice area (BA), white rice length (WL), white rice width (WW), white rice length-width ratio (WR) and white rice area (WA) are measured using a rice appearance quality detection analyzer (SC-E, Hangzhou Wanshen Test Technology Corporation, Hangzhou, China). Three technical repeats and two biological repeats were performed (average 120 seeds per plant), and the mean values were calculated for subsequent analysis.

Genotyping

DNA was extracted using the SDS lysis method (Zhang et al., 2019). The leaves of tillering stage were frozen with liquid nitrogen and ground with tissue grinder. SDS lysis buffer was added and incubated at 65 °C for 30 min. Sodium acetate was then added, the mixture was thoroughly mixed, and centrifuged. Isopropyl alcohol was added to the supernatant to precipitate the DNA, which was then rinsed twice with 70% ethanol and air-dried. The genotypic data were obtained using a 40K liquid-phase sequence chip (Li et al., 2022c), and all raw reads were filtered for high quality, with Q20 quality scores >95% and guanine-cytosine (GC) content <50%. Finally, 33,579 high-quality SNPs were obtained and used for genome-wide association study (GWAS) in this study. The genotypic data described here are accessible via FigShare (https://doi.org/10.6084/m9.figshare.26461429.v1).

Population genetic analysis

Principal component analysis (PCA) was conducted using the efficient mixed-model association (EMMA) method in the Genome Association and Prediction Integrated Tool (GAPIT) R package (Lipka et al., 2012) to examine the population structure. Kinship analysis of accessions is performed using the GAPIT package of R software and used for subsequent GWAS analysis. The K matrix (kinship matrix) was calculated by the EMMAX software (Kang et al., 2010) based on the Bayesian network (BN) method using those high-quality SNPs.

GWAS analyses

We performed a GWAS to detect SNPs that were significantly associated with all grain-shape-related traits using high-quality SNPs and the mean trait values of the 280 accessions. Marker-trait associations were conducted by the mixed linear model (MLM), PCA+K, implemented in the GAPIT package in R software (version 4.3.1) (Lipka et al., 2012). The critical P-value for declaring significant marker-trait association (1.0 × 10–4) was calculated using GEC software based on the independent effective SNP number (Li et al., 2012). To estimate independently associated regions of identified QTLs, significantly trait-associated SNPs situated in one estimated LD block were defined as the same QTL. Each LD block containing the identified SNPs was evaluated using the R package “LDheatmap” (Shin et al., 2006; Yano et al., 2016).

Identification of candidate genes

Haplotype analysis was conducted based on non-synonymous SNPs of QTL candidate genes in the temperate japonica rice of the 3K program (The 3,000 rice genomes project, 2014). Haplotypes containing more than 10 accessions were used to analyze significant differences in phenotype. Five representative candidate genes were selected for a comprehensive analysis based on the significance of the haplotype analyses (analysis of variance (ANOVA)), their biochemically related functions, and their expression profiles.

Statistical analysis

Differences in mean phenotypic values between haplotypes (consisting of more than 10 germplasm) were assessed using one-way ANOVA. Duncan’s multiple means comparison test uses the agricolae package in R software (4.3.1) to determine the significance of any difference (5% significance level). Phenotypic correlation analysis of grain-shape-related traits was calculated using the corrplot package in R. The variance components were evaluated using multi-site analysis, and all effects were treated as random.

Results

Phenotypic variation of grain-shape-related traits

The associations used in this study showed wide variations for grain-shape-related traits, and most traits appeared to be normally distributed (Table S1). Among them, the average GL, GW, and GR values of 280 accessions were 7.14, 3.12, and 2.32 mm, respectively (Figs. 1A–1C). Compared with grain, both brown rice and white rice exhibit decreases in both length and width to varying degrees. The mean values of BL, BW, and BR were 5.12, 2.81, and 1.83 mm, respectively (Figs. 1D–1F), while the average WL, WW, and WR values were only 4.84, 2.71, and 1.8 mm, respectively (Figs. 1H–1J). In addition, the average BA and WA values were 11.9 and 10.29, respectively (Figs. 1G, 1K). Correlation analysis showed that all 11 grain-shape-related traits were positively correlated, and the correlation coefficients between GR and BL, BW, and BR reached 0.99 (Fig. 1L).

Figure 1 Values and correlations of grain-shape-related traits.

(A–K) Box plots of grain length (A), grain width (B), grain length-width ratio (C), brown rice length (D), brown rice width (E), brown rice length-width ratio (F), brown rice area (G), white rice length (H), white rice width (I), white rice length-width ratio (J), and white rice area (K). GL, grain length; GW, grain width; GR, grain length-width ratio; BL, brown rice length; BW, brown rice width; BR, brown rice length-width ratio; BA, brown rice area; WL, white rice length; WW, white rice width; WR, white rice length-width ratio; WA, white rice area. (L) Correlations between the eleven tested traits. The areas and colors of ellipses correspond to the absolute values of the corresponding correlation coefficients. **Denotes significant correlations at P < 0.01.

Phylogenetic and population structure analysis

A total of 33,579 high-quality SNPs were obtained, and the average marker spacing was 11.1 kb. Among them, chromosome 1 had the most SNPs, containing 4,283, and the average marker spacing was 10.1 kb. Chromosome 9 contained the least number of markers, 1,986, with an average marker spacing of 11.5 kb (Fig. 2A, Table S2). Kinship and principal component analysis (PCA) were used to analyze the genetic structure of the SNPs in the 280 associations. The results showed that there were no obvious population structure groups among the 280 associations, because all of them came from northern China (Figs. 2B–2C).

Figure 2 Genotypic analysis.

(A) Heatmap illustrating SNP density in 280 accessions. (B) Heatmap of the kinship matrix, indicating the degree of relatedness between each pair of accession in the dataset. Darker colors indicate greater relatedness. (C) PCA of 280 accessions based on the screen plot within the rice diversity panel.

GWAS analysis of grain-shape-related traits

To identify genomic regions associated with the measured phenotypes, GWAS was performed on 280 associations. The Manhattan plot showed that when an association threshold of 1.0 × 10−4 was set, a total of 15 SNPs associated with grain shape were identified across the 280 associations, including one (qGL1), two (qGW5 and qGW6), one (qBA2), two (qBR5 and qBR6), two (qBW5 and qBW6), one (qWA2), two (qWR5 and qWR6), two (qWL2 and qWL7), and two (qWW5 and qWW6) for GL, GW, BA, BR, BW, WA, WR, WL, and WW, respectively (Table 1, Fig. S1). The identification of QTLs was achieved by defining contiguous SNPs exhibiting significant correlation and residing within an estimated linkage disequilibrium (LD) block. This approach yielded a total of five distinct QTL intervals for the 11 traits under study. Among these, qGL1 was found to influence GL, while qBA2 exhibited a multi-faceted effect, influencing BA, WL, and WA simultaneously. Additionally, two QTLs, qGW5 and qGW6 (despite being sequentially numbered, they both affect the same set of traits), were identified to have a combined influence on GW, BW, BR, WW, and WR. Lastly, a unique QTL, qWL7, was determined to specifically affect WL. These findings are summarized in Table 1.

Table 1 QTLs identified for grain-shape-related traits by GWAS.

Trait	QTL	Chr	Peak SNP	Interval (Mb)	P-value of peak SNPs	Previously reported QTLs and gene	
GL	qGL1	1	rs1_29399705	29.07–29.41	9.00439E−05		
GW	qGW5	5	rs5_5400007	5.30–5.46	1.22E−08	GW5 (Weng et al., 2008)	
	qGW6	6	rs6_26455250	26.40–26.68	4.09872E−05	GW6a (Song et al., 2015)	
BW	qBW5	5	rs5_5299885	5.30–5.46	2.98E−07	GW5	
	qBW6	6	rs6_26455250	26.40–26.68	7.31E−05	GW6a	
BR	qBR5	5	rs5_5460890	5.30–5.46	6.40E−09	GW5	
	qBR6	6	rs6_26455250	26.40–26.68	6.13E−08	GW6a	
BA	qBA2	2	rs2_33442570	33.04–33.50	2.27E−05	OsMKK4 (Duan et al., 2014)	
WL	qWL2	2	rs2_33357184	33.04–33.50	0.000037437	OsMKK4	
	qWL7	7	rs7_28430999	28.17–28.64	1.70739E−05	FZP (Ren et al., 2018)	
WW	qWW5	5	rs5_5460942	5.30–5.46	4.12117E−08	GW5	
	qWW6	6	rs6_26455146	26.40–26.68	1.70995E−06	GW6a	
WR	qWR5	5	rs5_5400006	5.30–5.46	2.59302E−09	GW5	
	qWR6	6	rs6_26455250	26.40–26.68	1.36456E−10	GW6a	
WA	qWA2	2	rs2_33442570	33.04–33.50	5.41E−06	OsMKK4	

Candidate gene identification

For the 11 grain-shape-related traits, we performed haplotype analysis to identify candidate genes for five QTLs consistently identified in five chromosomal regions (Table 2). Based on the Nipponbare reference genome IRGSP 1.0, the 29.07–29.41 Mb (336 kb) region containing qGL1 on chromosome 1 (Fig. 3A) harbored 53 candidate genes, among which four genes had known annotations in the Rice Annotation Project Database (RAP-DB, https://rapdb.dna.affrc.go.jp/). LOC_Os01g50720 encodes an OsMYB14 transcription factor and is a member of the rice MYB transcription factor gene family. The 3K program display that, a total of 2 SNPs were found in the coding region of MYB14 (Fig. 3B), forming two haplotypes, among which the GL of Hap1 was significantly higher than that of Hap2 (Fig. 3C). According to the published rice gene expression profile database (RiceXPro (version 3.0)), OsMYB14 is relatively highly expressed in specific organs (stem, lemma and palea) (Fig. 3D). Therefore, LOC_Os01g50720 is one of the most likely candidate genes in this region.

Table 2 List of the five most likely candidate genes for five quantitative trait loci (QTLs) associated with grain-shape-related traits.

QTL	Candidate gene	MSU_Locus_Annotation	Known function	
qGL1	LOC_Os01g50720	MYB family transcription factor, putative, expressed		
qBA2, qWL2, qWA2	LOC_Os02g54600(OsMKK4)	STE_MEK_ste7_MAP2K.5–STE kinases include homologs to sterile 7, sterile 11 and sterile 20 from yeast, expressed	Influences grain size in rice	
qGW5, qBW5, qBR5, qWW5, qWR5	LOC_Os05g09520(GW5)	IQ calmodulin-binding motif family protein, expressed	Regulate grain width in rice	
qGW6, qBW6, qBR6, qWW6, qWR6	LOC_Os06g44100(GW6a)	HLS, putative, expressed	Regulate grain width in rice	
qWL7	LOC_Os07g47330(FZP)	AP2 domain containing protein, expressed	Influences grain size in rice	

Figure 3 Candidate genes analysis of qGL1 on chromosome 1.

(A) Linkage disequilibrium (LD) block surrounding the peak on chromosome 1. (B) Gene structure of LOC_Os01g50720 and DNA polymorphisms within that gene. Blue boxes, white boxes and straight lines represent exons, introns and untranslated regions (UTR), respectively, and arrows represent gene directions. (C) Boxplots illustrating grain length based on haplotypes for LOC_Os01g50720 using non-synonymous single nucleotide polymorphisms (SNPs) within the coding region. **Denotes the significance of ANOVA at P < 0.01. (D) Spatio-temporal expression patterns of LOC_Os01g50720 in various Nipponbare tissues throughout the entire growth period in the field (downloaded from RiceXPro (version 3.0)).

There are 69 candidate genes in the 33.04–33.50 Mb (462 kb) region of qBA2 on chromosome 2, among which six genes have known annotations in the RAP-DB (Fig. 4B). One of these candidate genes, LOC_Os02g54600, is identical to OsMMK4, a previously reported small grain gene that regulates grain shape (Duan et al., 2014; Xu et al., 2018). The results of haplotype analysis showed that the coding region of OsMMK4 contained 2 SNPs (Fig. 4A), forming two haplotypes, and the particles carrying Haplotype Hap1 materials were significantly higher than those carrying Hap2 materials (Fig. 4C). OsMMK4 encodes a mitogen-activated protein kinase that influences rice grain shape by regulating cell proliferation, so we predict that this gene is a candidate for the qBA2 region.

Figure 4 Candidate genes analysis of qBA2 on chromosome 2.

(A) Gene structure of LOC_Os02g54600 and DNA polymorphisms within that gene. Blue boxes, white boxes and straight lines represent exons, introns and untranslated regions (UTR), respectively, and arrows represent gene directions. (B) Linkage disequilibrium (LD) block surrounding the peak on chromosome 2. (C) Boxplots illustrating grain length based on haplotypes for LOC_Os02g54600 using non-synonymous SNPs within the coding region. **Denotes the significance of ANOVA at P < 0.01.

For GW, BW, BR, WW and WR, a high peak of qGW5 on chromosome 5 was mapped together with the peak of qBW5, qBR5, qWW5, qWR5. A total of 20 annotated genes were selected from the region of 5.30–5.46 Mb (161 kb) (Fig. 5A). Among them, LOC_Os05g09520 is identical to Grain Width on chromosome 5 (GW5), which regulates the expression level and growth response of brassinolide (BR) response genes (Liu et al., 2017). The coding region of LOC_Os05g09520 contains a total of three SNPs and divides the material into two haplotypes (Fig. 5B). Hap2 showed significantly larger GL and GR values than Hap1 and showed significantly lower GW values than Hap1 (Figs. 5C–5E). Therefore, GW5 is the most likely candidate gene in this region.

Figure 5 Candidate genes analysis of qGW5 on chromosome 5.

Candidate genes analysis of qGW5 on chromosome 5. (A) Linkage disequilibrium (LD) block surrounding the peak on chromosome 5. (B) Gene structure of LOC_Os05g09520 and DNA polymorphisms within that gene. Blue boxes, white boxes and straight lines represent exons, introns and untranslated regions (UTR), respectively, and arrows represent gene directions. (C–E) Boxplots illustrating grain length (C), grain width (D) and grain length-width ratio (E), respectively, based on haplotypes for LOC_Os05g09520 using non-synonymous SNPs within the coding region. **Denotes the significance of ANOVA at P < 0.01.

A QTL cluster (qGW6, qBW6, qBR6, qWW6 and qWR6) affecting GW was identified in the region of 26.40-26.68 Mb (279 kb) on chromosome 6, containing 40 annotated genes (Fig. 6B). Where, LOC_Os06g44100 encoding a GNAT-like protein, that harbors intrinsic histone acetyltransferase activity (OsglHAT1) (Song et al., 2015). No SNPs were identified in the coding region of LOC_Os06g44100, but two SNPS were found in the promoter region, dividing all materials into two haplotypes (Fig. 6A). Hap1 exhibited a significantly higher GW value than Hap2 haplotypes in the whole population (Fig. 6C).

Figure 6 Candidate genes analysis of qGW6 on chromosome 6.

(A) Gene structure of LOC_Os06g44100 and DNA polymorphisms within that gene. Blue boxes, white boxes and straight lines represent exons, introns and untranslated regions (UTR), respectively, and arrows represent gene directions. (B) Linkage disequilibrium (LD) block surrounding the peak on chromosome 6. (C) Boxplots illustrating grain length based on haplotypes for LOC_Os06g44100 using SNPs within the promoter region. **Denotes the significance of ANOVA at P < 0.01.

In addition, qWL7 was identified in the region of 28.17–28.64 Mb (472 kb) on chromosome 7 (Fig. 7B), which contains 84 annotated genes. Among them, LOC_Os07g47330 encodes a ERF domain protein, and mutations in LOC_Os07g47330 result in small grain and dense panicle phenotypes (Komatsu et al., 2003; Bai et al., 2017; Ren et al., 2018). LOC_Os07g47330 has only one SNP in its coding region, dividing the material into two haplotypes (Fig. 7A). The GL value of Hap1 is significantly lower than that of Hap2 (Fig. 7C).

Figure 7 Candidate genes analysis of qWL7 on chromosome 7.

(A) Linkage disequilibrium (LD) block surrounding the peak on chromosome 7. Blue boxes, white boxes and straight lines represent exons, introns and untranslated regions (UTR), respectively, and arrows represent gene directions. (B) Gene structure of LOC_Os07g47330 and DNA polymorphisms within that gene. (C) Boxplots illustrating grain length based on haplotypes for LOC_Os07g47330 using non-synonymous SNPs within the coding region. **Denotes the significance of ANOVA at P < 0.01.

Discussion

Rice grain shape is a crucial agronomic trait that significantly impacts rice yield and quality. Although recent studies have identified some key regulators of grain size and molecular regulatory pathways, the understanding of the entire regulatory network remains limited and fragmented. Therefore, it is necessary to explore the upstream and downstream components of the known regulators of grain size and the possible links between the identified regulatory pathways. In this study, we conducted a genome-wide association study (GWAS) on 280 japonica rice varieties from northern China to decipher the genetic basis of grain shape traits. By measuring 11 grain-related traits and performing high-density single nucleotide polymorphism (SNP) genotyping, we identified 15 QTLs associated with these traits. Notably, five major QTL clusters emerged, revealing key candidate genes such as LOC_Os01g50720, OsMKK4, GW5, GW6a, and FZP. In comparison to previous studies, our GWAS analysis provided a more comprehensive understanding of rice grain shape by including indicators related to both brown and white rice shapes. This approach allowed us to gain insights into the interplay between glume development, grain filling, and endosperm development. For instance, we identified QTLs that specifically influence brown rice width (qBW5, qBW6) and white rice width (qWW5, qWW6), in addition to overall grain width (qGW5, qGW6). These findings contribute to a more nuanced understanding of the genetic control of rice grain shape. Our findings provide valuable insights into the genetic mechanisms underlying rice grain shape and suggest potential targets for marker-assisted selection to enhance rice quality and yield.

Recent research has revealed that rice grain shape is predominantly regulated by G-protein signaling, mitogen-activated protein kinase (MAPK) pathways, the ubiquitin-proteasome system, phytohormone signaling, and transcriptional regulators (Zuo & Li, 2014; Li, Xu & Li, 2019; Ren, Ding & Qian, 2023). Among the five grain shape-related QTLs identified in this study, four are located within the same or adjacent regions as previously reported QTLs and cloned genes influencing rice grain shape traits. For instance, we identified GW5 as a significant candidate gene for controlling grain width and related traits in our study. GW5 encodes a nuclear localization protein that inhibits the activity of glycogen synthase kinase GSK2, thereby regulating the expression level of brassinolide response genes and grain growth (Weng et al., 2008; Liu et al., 2017). The consistent identification of GW5 across different studies underscores its pivotal role in determining grain width in rice. our study reinforces this notion by concurrently identifying the influence of GW5 on grain width, brown rice width, and milled rice width, highlighting the reliability of our phenotypic data and emphasizing the pivotal role of GW5 in regulating grain shape in japonica rice cultivars from Northern China. In addition to GW5, we also identified OsMKK4 as a candidate gene for influencing multiple grain shape traits. OsMKK4 encodes a mitogen-activated protein kinase kinase that has been previously reported to regulate grain shape by modulating brassinosteroid (BR) responses (Xu et al., 2018). Our haplotype analysis revealed two distinct haplotypes of OsMKK4, with Hap2 showing significantly higher grain length values compared to Hap1. This finding suggests that variations in OsMKK4 may contribute to differences in grain shape among rice varieties, providing new avenues for rice breeding through the manipulation of OsMKK4 alleles. Furthermore, in contrast to previous studies that did not identify different genotypes of OsMKK4 in natural materials, we have successfully identified 13 elite germplasm lines harboring a novel haplotype of this gene. Another notable candidate gene identified in our study was GW6a, which is associated with grain width and related traits. Previous studies have primarily linked GW6a with the regulation of grain weight through changes in grain length (Song et al., 2015). However, our haplotype analysis indicated that GW6a may also influence grain width, which may be attributed to variations in the genetic backgrounds and environmental conditions of the materials used in different studies. Mutation of FZP causes smaller grains and degenerated sterile lemmas. Two haplotypes of LOC_Os07g47330 were detected, with Hap 2 being associated with a significantly larger GL value than Hap 1 (Fig. 6C), suggesting that LOC_Os07g47330 is a likely candidate gene for qWL7. However, current research on the function of the FZP gene primarily focuses on its role in regulating the formation of axillary bud meristems, while there is limited understanding of its molecular mechanism in regulating rice grain shape. Furthermore, we identified LOC_Os01g50720 as a candidate gene for influencing grain length. LOC_Os01g50720 encodes an OsMYB14 transcription factor, which belongs to the MYB transcription factor gene family in rice. Although MYB family genes have primarily been associated with responses to non-biotic stresses (Kang et al., 2022), the presence of plant hormone-related response elements in these genes suggests that OsMYB14 may play a role in regulating grain development. Further functional validation is needed to confirm the role of OsMYB14 in rice grain shape. Despite the similarities, our study also revealed some differences compared to previous research. One possible reason for these differences could be the genetic diversity of rice varieties used in our study. Our panel consisted of 280 japonica rice varieties from northern China, which may have unique genetic variations compared to other rice germplasm resources. Additionally, environmental factors such as climate, soil type, and cultural practices can also influence the expression of grain shape traits, leading to variations in QTL detection across different studies. In conclusion, our GWAS analysis of japonica rice varieties from northern China provided valuable insights into the genetic mechanisms underlying rice grain shape.

Rice yield is a complex quantitative trait controlled by multiple factors and genes, with grain shape, including grain length, grain width, grain thickness, and length-to-width ratio, being one of the crucial determinants of grain weight (Sakamoto et al., 2006). Generally, rice milling quality is negatively correlated with grain length, length-to-width ratio, and length-to-thickness ratio, while it is positively correlated with grain width, grain thickness, and width-to-thickness ratio (Meng et al., 2022). Additionally, rapid grain filling in wide-grain rice can result in a looser arrangement of starch granules, leading to the formation of chalkiness, which negatively impacts the appearance and milling quality of rice (Li et al., 2022b). Therefore, selecting new varieties of narrow-grain, high-quality rice, reducing grain width to enhance rice quality, and compensating for any loss in grain weight by increasing grain length, can improve rice quality without sacrificing yield. In this study, we identified three accessions, including LX12, LG237, and SN159, that carry low-GW haplotypes of GW5 at qGW5, qBW5, qBR5, qWW5, and qWR5, as well as GW6a at qGW6, qBW6, qBR6, qWW6, and qWR6. Two accessions, YG752 and MF9, carry high-GL haplotypes of LOC_Os01g50720 at qGL1 and FZP at qWL7. Although these candidates are not causal genes, the SNPs in their sequences are suitable for marker-assisted selection (MAS) due to the high degree of linkage disequilibrium between them. Consequently, by converting these linked SNPs into Kompetitive Allele Specific PCR (KASP) markers, MAS can be implemented to improve grain shape, potentially enhancing rice quality through the introgression of low-GW alleles (haplotypes) of GW5 and GW6a, as well as high-GL alleles of LOC_Os01g50720 and FZP into high-yielding varieties. Our findings suggest potential targets for MAS in rice breeding programs aimed at enhancing both grain quality and yield. Further functional validation of the identified candidate genes and exploration of their interactions within the genetic network will be essential for advancing our understanding of rice grain shape regulation.

Conclusions

In conclusion, our genome-wide association study of 280 japonica rice varieties from northern China uncovered crucial quantitative trait loci (QTLs) influencing grain shape traits. By conducting extensive haplotype analyses and leveraging functional annotations, we identified five major QTL clusters associated with grain length, width, and area, pinpointing key candidate genes such as LOC_Os01g50720, OsMKK4, GW5, GW6a, and FZP. These findings not only advance our understanding of the genetic architecture underlying rice grain shape but also provide valuable genetic markers for precision breeding aimed at enhancing rice quality and yield. The implementation of marker-assisted selection strategies incorporating these loci and genes holds significant promise for improving rice varieties and contributing to global food security.

Supplemental Information

Supplemental Information 1 Genome-wide association results for grain-shape-related traits.

Manhattan plots (left) and quantile-quantile plots (right) associated with GL, GW, GR, BL, BW, BR, WL, WW and WR in 280 accessions. For the Manhattan plots, -log10 P-values from a genome-wide scan were plotted against the position of the SNPs on each of 12 chromosomes and the horizontal red lines show the suggestive threshold P = 1.0 × 10−4. For the quantile-quantile plots, the horizontal axes indicate the −log10-transformed expected P values, and the vertical axes indicate the −log10-transformed observed P-values.

Supplemental Information 2 Summary of 280 rice accessions and the grain-shape-related traits.

Supplemental Information 3 The distribution of single nucleotide polymorphism (SNP) markers on chromosomes.

Additional Information and Declarations

Competing Interests

Author Contributions

DNA Deposition

Data Availability

The authors declare that they have no competing interests.

Hongwei Chen conceived and designed the experiments, performed the experiments, analyzed the data, prepared figures and/or tables, authored or reviewed drafts of the article, and approved the final draft.

Xue Zhang performed the experiments, prepared figures and/or tables, and approved the final draft.

Shujun Tian performed the experiments, prepared figures and/or tables, and approved the final draft.

Hong Gao performed the experiments, authored or reviewed drafts of the article, and approved the final draft.

Jian Sun analyzed the data, authored or reviewed drafts of the article, and approved the final draft.

Xiu Pang performed the experiments, prepared figures and/or tables, and approved the final draft.

Xiaowan Li performed the experiments, prepared figures and/or tables, and approved the final draft.

Quanying Li performed the experiments, prepared figures and/or tables, and approved the final draft.

Wenxiao Xie performed the experiments, prepared figures and/or tables, and approved the final draft.

Lili Wang performed the experiments, authored or reviewed drafts of the article, and approved the final draft.

Chengwei Liang analyzed the data, prepared figures and/or tables, and approved the final draft.

Guomin Sui conceived and designed the experiments, authored or reviewed drafts of the article, and approved the final draft.

Wenjing Zheng conceived and designed the experiments, authored or reviewed drafts of the article, and approved the final draft.

Zuobin Ma conceived and designed the experiments, authored or reviewed drafts of the article, and approved the final draft.

The following information was supplied regarding the deposition of DNA sequences:

The genotypes described here are available at FigShare: Chen, Hongwei (2024). Genotype of japonica accessions from northern China. figshare. Dataset. https://doi.org/10.6084/m9.figshare.26461429.v1.

The following information was supplied regarding data availability:

The raw measurements are available in the Supplemental File.

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
