# Peer review of "Genome-wide association study reveals the advantaged genes regulating japonica rice grain shape traits in northern China"

_PeerJ, doi:10.7717/peerj.18746_

## Round 0.1 · original submission · Major Revisions

Your manuscript requires major revisions that include comments from reviewers.

Reviewer 1 ·

Basic reporting

The manuscript presents a genome-wide association study (GWAS) aimed at identifying genes linked to grain shape traits in Japonica rice varieties in northern China. While the key findings are promising and could potentially contribute to agricultural genomics and rice breeding programs, several critical issues need to be addressed before the manuscript can meet the scientific rigor required for publication in PeerJ.
There are several typos and grammatical errors throughout the manuscript that reduce the overall readability and coherence. A thorough proofreading and copy-editing effort is necessary to correct these issues. Beyond grammatical improvements, the manuscript should also focus on improving clarity, particularly in the methodology, results and discussions. The scientific arguments need to flow logically and be presented in a more structured manner. Please consider engaging a professional editing service or a native English-speaking co-author for revision.
The introduction lacks sufficient context regarding the genetic and agronomic importance of grain shape traits in Japonica rice. While the authors focus heavily on previously studied genes and QTLs controlling grain traits, they fail to adequately introduce the novelty of their study. This leaves a gap in understanding how their work builds upon or advances prior knowledge. A clear research rationale, hypothesis, and research question are missing, which makes it difficult to grasp the core objective of the study. The introduction should provide a more comprehensive review of the literature, especially recent advancements in GWAS related to rice grain traits. Clearly state how the study’s focus on northern China’s Japonica varieties is distinct or innovative. Please introduce specific research hypotheses, outline the scientific questions addressed by the study, and discuss how these questions align with the methods and results.
The abstract is overly general and lacks specific data points. It should summarize the key findings quantitatively, including important metrics such as the number of significant associations identified, key QTLs, or candidate genes, and any quantitative improvements in grain shape traits. Additionally, the conclusion in the abstract should not be vague but should provide a concise summary of the study’s contributions. Please, add key numerical values to the abstract, succinctly summarize the results, and provide a clear, brief conclusion.
The Materials and Methods section lacks sufficient detail for reproducibility, a key criterion for publication.
Quality control procedures for genotypic and phenotypic data,
Details on experimental setup (e.g., environmental conditions for growing rice varieties).
These are either missing or insufficiently explained. Without this information, it is difficult for other researchers to replicate or build upon the study. The inclusion of software versions, algorithms used for GWAS analysis, and specifics on any applied corrections for population stratification (such as PCA or kinship models) would greatly enhance the rigor of the methods section. Please, expand the methods to provide full transparency and ensure reproducibility. The addition of supplementary material, if necessary, could aid in providing the detailed methodological framework.
The results section is unclear and difficult to follow. The presentation of results should be improved both stylistically and structurally. In its current form, it lacks coherence and does not effectively highlight the most significant findings. The organization of the section makes it challenging to differentiate between major and minor findings.
Please reorganize the results section for a better logical flow, and ensure that every figure and table is clearly referenced and discussed in the text. Summarize key findings before delving into minor details.
The Discussion section requires a more in-depth interpretation of the results, particularly in the context of existing literature. The authors should:

• Elaborate on the significance of the identified genes and their potential role in grain shape traits.
• Discuss the implications of these findings for rice breeding programs in northern China.
• Compare the results with other studies in the field and explain any similarities or discrepancies with previously identified QTLs.
• Additionally, there should be a more critical assessment of the limitations of the study, including any potential biases in the GWAS approach used or limitations in sample size and environmental variation.
Please expand the discussion to incorporate a broader comparison with recent studies and offer mechanistic interpretations of the identified associations.
The Conclusion is currently a near-repetition of the results and lacks the broader generalizations necessary to make the findings impactful. Rather than merely restating results, the conclusion should:
• Address the research hypothesis and questions posed in the introduction.
• Discuss the broader implications of the findings for rice breeding and genetics.
• Highlight future directions for research, including potential functional studies of identified genes or the application of findings in crop improvement.
• Recommendation: Focus on the mechanistic linkage of findings and implications for the broader research field or practical applications.
Figures and Tables: Ensure that all figures and tables are well-labeled, referenced appropriately, and contribute meaningfully to the results section. Poor-quality or unclear figures should be revised to enhance clarity.
References: It is critical to ensure that the most recent and relevant literature is cited, particularly in the introduction and discussion.
Specific comments:
Please check the first line of Introduction in Line No. 44: “The genetic mechanism of rice grain shape is a key field in the study of rice quality and yield(Li et al., 2022a).” There is a missing space between text and citation and this mistake is repeated throughout the manuscript.
Line No. 115 “After the water content of grain was stabilized” Please clearly mention how the water content was stabilized?
Line No. 122: “DNA was extracted by SDS cracking method.” please provide the reference of SDS protocol used for DNA extraction.
Line No. 122-126 “First, 8 pieces of leaves were taken with a hole punch and placed in a sampling tray, steel balls were added, frozen at -80 C, the leaves were ground with a tissue grinder and then added with 1.5% SDS cracking solution, incubated at 65 for cracking, then centrifuged with sodium acetate to obtain supernatant, isopropyl alcohol was added and DNA was precipitated, rinsed with 70% alcohol twice and let dry to get clean DNA.
Please break the sentence into simpler sentences. The statement of pieces of leaf via hole punch is not clear, Make it clear. At which stage leaf were sampled for DNA extraction, please mention. What was the purpose of steel ball addition?
Line No. 126: “DNA was precipitated”. Please use appropriate terminology.
Line No. 127: “The genotypes were obtained using a 40K liquid-phase sequence” Use proper expression and language. Is it genotypic data i.e., sequencing of DNA extracted?
Line No. 129 “Finally, 33,579 high-quality SNPs were selected” Clearly mention the selection criteria of SNPs. There are so many SNPs, which one you used? At least mention or provide some data.
Please provide source of SNPs and also for locus annotation. Source/databases?
Data of genotyping or sequencing is not provided. Which databases were used? Please clearly mention materials and method section. Rather than providing in later sections.
No reference is given for Kinship matrix?
Please re-check the interpretation of Table 1 in Line 192-195.
Figure 2C, please provide biplot of PCA analysis presenting the relation of traits and genotypes both.
In Figure 3 nothing highlighted in figure, neither any distance nor region specification labelling.
Please provide link of Rice databases.
How and through which database haplotypes coding regions were identified in MYB14 i.e., HAP1 and HAP2, please provide source link etc.
In table 1 please provide the references of previously reported QTLs and genes.
In table 2: What is IQ calmodulin-binding motif. Explain all the abbreviations in the footnotes of tables.
Recheck the interpretation of Fig. 5C-E. It should have been reverse for given results.
Fig. 7 why was HAP1’s value not mentioned in the figure while given in the text.
Please label B diagram of all the figures for annotated genes or Haplotypes. Make it clear what color represents and what triangles are indicating.

Experimental design

Experimental design is good but needs more clarifications for enhancing its overall quality.

Validity of the findings

The findings seems valid and promising for rice crop breeding.

Additional comments

The article is sound practically and statistically, provided the authors add the missing information as mentioned in basic reporting.

Reviewer 2 ·

Basic reporting

The authors studied “Genome-wide association analysis reveals the advantaged genes of japonica grain shape traits in northern China” which is an important aspect of rice Breeding. The study used 280 japonicum rice varieties and reported 15 QTLS for Grain shape (i.e. length, width) before and after Shelling/milling during 2021. However, a few concerns must be addressed before the article can meet the standards of the journal.

First of All, the article needs lots of language cleanup. Many sentences are written that do not follow the sentence structure in English, especially, in the Materials and Methods and Results section.
Secondly, the abstract should follow the Journal instructions for Abstract writing. It contains headings and different structures, which seems absurd.
Thirdly, in the introduction, the authors should clearly explain the criteria and the importance of grain length and width criteria, explicitly outlining the ideal characteristic features of Japonica rice grains. This includes defining the desired traits, such as grain length, width, shape, and size, which are crucial for determining grain quality. By doing so, the authors can clearly articulate the study's objectives, focusing on the specific aspects of grain characteristics they aim to improve.
A comprehensive explanation of the transcription factors' roles and potential mechanisms, bridging the gap between genetic regulation and phenotypic expression is explained but it is not clear, how those events relate to this investigation.

I suggest, the authors should re-write the introduction part by defining the standard criteria for selecting grain length and width, including the acceptable ranges and thresholds.
They should describe the ideal characteristic features of Japonica rice grains, such as grain shape, size, and appearance.
Inclusion of discussion about the factors related to grain shape, including their roles, functions, and potential interactions should be added.
By expanding the introduction in this way, the authors can provide a more comprehensive and nuanced exploration of the topic, setting the stage for a well-structured and informative study.

Experimental design

The experiment's reliance on one-year, one-location data (2021) may lead to skewed results, as environmental factors can significantly impact the outcome in a single year. To ensure more robust and reliable findings, it is suggested to consider data from multiple years (e.g., 2022, 2023) and locations should be added if available.
This approach will minimize the impact of annual variability in weather patterns, soil conditions, and other environmental factors. The results drawn will have better validity and increased confidence in the conclusions drawn from the study, as they are based on a more extensive and diverse dataset.

For field studies, it is generally recommended to collect data over at least two years to account for any skewed effects of the environment in a particular year.

While writing the background of the study the authors wrote “The cloning of rice grain shape related gene and its related regulatory pathway network need further study”
The authors should rephrase it by deciphering the two parts very clearly i.e. in the first part the QTL characterization and in the second part their exploitation in a breeding program.
Well, the characterization of the genes/QTLs and their intricate pathways can help to achieve the objectives laid out in the study.

Validity of the findings

It is unclear: what was the basis of the selection of those 280 japonica rice varieties? Were they randomly selected or they were elite cultivars for rice grain shape and milling quality?
Line 28-31, the authors wrote “280 japonica rice varieties from northern China were used to conduct genome-wide association analysis for grain-shape-related traits, and to explore the relevant loci or genes controlling grain shape.”
Of the 15 reported QTLs, 14 were already reported. How is this study unique and how much it contributes to the Knowledge?

The authors fail to report the phenotypic contribution of the QTL in total variation for the trait. This means the next step, where it is decided which QTL should be used in the breeding program is missing. This makes this study incomplete.

Line 31-32, the authors wrote “A total of 15 QTL affecting the eleven traits were detected using 33,579 high-quality single nucleotide polymorphism markers. ”

What is meant by “high Quality” SNP markers?
It is not mentioned whether 90K illumine Chip was used or DArT markers were assayed or any other high throughput technique was used. And how those 33,579 markers were eventually selected (i.e. based on their reproducibility scores? Or any other criterion).

The authors wrote in line 34-37 “Among them, LOC_Os01g50720 for qGL1; OsMKK4 (LOC_Os02g54600) for qBA2, qWL2 and qWA2, GW5 (LOC_Os05g09520) for qGW5, qBW5, qBR5, qWW5 and qWR5; GW6a (LOC_Os06g44100) for qGW6, qBW6, qBR6, qWW6 and qWR6; FZP (LOC_Os07g47330) for qWL7 were considered as the most likely candidate genes based on functional annotations and haplotype analysis.”

The QTL naming should follow the guidelines for the Gene Nomenclature system for Rice which states
"Rice QTL nomenclature rules [14] indicate that each QTL name should be italicized and start with a lower case letter “q” to indicate that it is a QTL, followed by a two to five letter standardized “trait name” (e.g. SW for Seed Width), a number designating the rice chromosome on which it occurs (1–12), a period (“.”), and a unique identifier to differentiate individual QTLs for the same trait that reside on the same chromosome (e.g. qSW5.1). When QTLs are entered into a genome database such as Gramene [9], they may be further assigned a standardized trait term from the Trait Ontology (TO; [10]; e.g. seed width, Accession #TO: 0000140) to facilitate querying and may be assigned a new, unique identifier to avoid confusion between studies. In any case, this database assignment will be reflected as a synonym within the QTL record, and the original, published QTL name will be retained for search purposes.”
The rice QTL nomenclature should be followed throughout the article with a uniform style.

The authors studied in Line 115-120
“After the water content of grain was stabilized, about 200 grains were selected and grain length(GL), grain width(GW), grain length-width ratio(GR), brown rice length(BL), brown rice width(BW), brown rice length-width ratio(BR), brown rice area(BA), white rice length(WL), white rice width(WW), white rice length-width ratio(WR) and white rice area(WA) were measured by rice appearance quality detection analyzer. The mean was calculated for subsequent analysis”.

It is not clearly explained: what is difference between Grain Length, Brown Rice Length, and White rice Length. The Materials and Methods section should write about shelling process in rice, and its importance about the yield losses before and after milling.

The authors wrote in line 122-127 “DNA was extracted by SDS cracking method. First, 8 pieces of leaves were taken with a hole punch and placed in a sampling tray, steel balls were added, frozen at -80 ℃, the leaves were ground with a tissue grinder and then added with 1.5% SDS cracking solution, incubated at 65 ℃ for cracking, then centrifuged with sodium acetate to obtain supernatant, isopropyl alcohol was added and DNA was precipitated, rinsed with 70% alcohol twice and let dry to get clean DNA”

It looks like it is translated from any other language as “SDS lysis buffer” is translated in “SDS Cracking buffer” which does not make any sense, and should be replaced.

Additional comments

Tables and figures are missing foot note etc.Tables and Figures should be presented in a clear and self-explanatory manner, allowing readers to understand the data without referencing the text. To achieve this, each table and figure should Have a concise and descriptive title, accurately summarizing the content and facilitating quick comprehension.
Include a footnote or legend, providing essential context and explanations for the data, such as:
- Definitions of abbreviations and symbols used
- Explanation of data collection methods or sources
- Clarification of unusual or complex data points
- Statistical significance or probability values (e.g., p-values)
Be designed to stand alone, allowing readers to interpret the results without needing to refer to the text for explanation.
Be numbered and referenced in the text, enabling easy navigation and cross-referencing.
Adhere to a consistent format and style throughout the study, enhancing readability and visual appeal.

Reviewer 3 ·

Basic reporting

1. The superscript numbers of the authors are written incorrectly. They should follow a certain order. Also, an author cannot serve in two places at the same time. The parts labeled with 1,2 need to be corrected.
2. The article is about rice, but there isn't a single word related to paddy in the title.
3. Line 44. The genetic mechanism of rice grain shape is a key field in the study of rice quality and yield(Li.
4. I suggest that the introduction starts with information about rice, and then gradually and smoothly transitions to the main topic
5. Line47-51. There is no citation, give a reference, please.
6. Between all parenthesis please put space. And to this for an entire manuscript.
7. Line 52-91. Too long a paragraph. Dive it, please.
8. Line-98-99. Situ et al., it is without year. Pay attention.
9. Line 122. Please give reference or details. (DNA was extracted by SDS cracking method).
10. There is no space between passages.
11. Where did you do this analysis, ı mean lab.?

Experimental design

-

Validity of the findings

-

Additional comments

The discussion section is very weak. It is almost no different from the conclusion. The article is well-constructed and has had a lot of effort put into it, but there are aspects that require major revisions. In the discussion section, the findings should be compared with the literature, and comments should be made about the differences and similarities. Ideas about the reasons for these differences should be proposed. Additionally, the genes mentioned in the paragraph between Line 52 and Line 91 are irrelevant to the text, so providing detailed information about them is unnecessary. Also, there is no word related to paddy in the title

---

## Round 0.2 · Major Revisions

It appears that you have not yet made enough progress in your manuscript to address the reviewers' concerns.

Reviewer 1 ·

Basic reporting

The revised version of the manuscript presents a well-structured and comprehensive study. Overall, only minor changes during proofreading will be needed to prepare this manuscript for publication.
Several sentences could benefit from simplification. For example, the sentence “At present, many genes or QTLs related to grain length in rice have been cloned, including qGL3, SMOS1, GS3, GS2, PGL1, TGW6, PGL2, GL7 and so on...” could be more direct: “Many genes and QTLs controlling grain length have been cloned, such as qGL3, SMOS1, GS3, and others.”
Please ensure verb tense consistency like in Line No. 122, The sentence “When the accessions are mature, 5 plants are selected...” should use past tense to match the rest of the section: "When the accessions were mature, five plants were selected..."
Line No. 122, “5 plants” should be written as “five plants” to maintain consistency in formal writing.
The mention of a “rice appearance quality detection analyzer” is vague. The specific make and model should be provided to enhance reproducibility.

Line No. 179-180: Compared with grain, the relevant indicators of brown rice and white rice all declined to varying degrees. Clarify this sentence.

In line 216, replace "3K program display" with "The 3K program displays that..." or "According to the 3K program, two SNPs were found in...". The current phrasing is awkward.

Experimental design

The experimental design is compelling, utilizing a robust genome-wide association analysis (GWAS) with a substantial sample size of 280 rice accessions, which provides a solid foundation for detecting significant quantitative trait loci (QTLs) through adequate genetic diversity. The integration of SNP genotyping and haplotype analysis enhances the rigor of the study, facilitating the reliable identification of candidate genes associated with grain shape traits.

Validity of the findings

The findings of the study are valid, as they are supported by a comprehensive analysis of genetic diversity, significant associations between identified QTLs and grain shape traits, and the functional characterization of candidate genes, all of which contribute to a deeper understanding of the genetic architecture underlying rice grain morphology.

Reviewer 2 ·

Basic reporting

The article is improved in terms of Language cleanup now, as many sentences have been re-written.
However, my comment one of my comments is not addressed:

The authors were supposed to write about the standard criteria that can define a rice grain length, width, length-width ratio, etc. It is still pending.

It is imperative to explain, as it gives readers a standard shape that is being aimed at while breeding for improvement in shape-related traits.

Experimental design

It is unclear how many plants and grains were taken for each trait if we can not have multiple years and multiple location data. Please explicitly report how many samples were taken from each plot and how many grains were observed from each plant (to make it pseudo-replicate).

Validity of the findings

The QTL naming still seems to be an issue, and once reported it can cause confusion.
Can authors name all the QTLs reported in a similar fashion?

Reviewer 3 ·

Basic reporting

The author did not enrich the discussion section. If I were asked, I would say that the most important part of a paper, the section that contributes the most to science, is the discussion section. The comparisons, similarities, differences, and the interpretations of the reasons behind these elements shed light on science. However, in this paper, such comparisons are not present. I am in favor of sending the paper back to the author for strengthening the discussion section. The author does not seem to take the reviewers' comments seriously. Under normal circumstances, I would have rejected this, but I believe it deserves another chance because it is an original work with a lot of effort put into it.

Experimental design

-

Validity of the findings

-

Additional comments

-

---

## Round 0.3 · Major Revisions

Because you have not followed Reviewer 3's suggestion to improve the discussion, I am submitting your manuscript for major revision again. You should carry out the revision by taking into account the comments of all of the reviewers, not just one of them, and write answers to them.

Reviewer 2 ·

Basic reporting

The submitted article has undergone careful revision from the authors and is deemed to be in good shape, meeting the standard criteria of the journal.

Experimental design

Looks Ok. and should be allowed to proceed to publish.

Validity of the findings

Seems Ok

Reviewer 3 ·

Basic reporting

Hello dear author(s). The Discussion section of the article is the most important part, as it contributes the most to science, guides the field, and sheds light for other researchers, enhancing their ability to think and interpret. I have now requested a major revision twice because this section has not been strengthened. Please read at least 50 articles, focusing specifically on the Discussion sections, and enrich your article accordingly. A well-constructed Discussion would suit a study of this caliber. Additionally, I recommend making your responses to the reviewers more systematic. Address each comment point-by-point, and if the reviewers' comments contradict your work, please provide the necessary defense in your response to them.

Experimental design

-

Validity of the findings

-

Additional comments

-

---

## Round 0.4 · accepted · Accept

The changes you made were considered sufficient by the reviewers. Your corrections are sufficient for me to accept your manuscript. Congratulations

Reviewer 3 ·

Basic reporting

From my side it is ok. The corresponding author did the required corrections.

Experimental design

-

Validity of the findings

-

Additional comments

-